

# Pylogeny: an open-source Python framework for phylogenetic tree reconstruction and search space heuristics

Alexander Safatli[1] and Christian Blouin[1,2]

[1] Faculty of Computer Science, Dalhousie University, Halifax, NS, Canada
[2] Department of Biochemistry and Molecular Biology, Dalhousie University, Halifax, NS, Canada

## ABSTRACT

**Summary.** Pylogeny is a cross-platform library for the Python programming language that provides an object-oriented application programming interface for phylogenetic heuristic searches. Its primary function is to permit both heuristic search and analysis of the phylogenetic tree search space, as well as to enable the design of novel algorithms to search this space. To this end, the framework supports the structural manipulation of phylogenetic trees, in particular using rearrangement operators such as NNI, SPR, and TBR, the scoring of trees using parsimony and likelihood methods, the construction of a tree search space graph, and the programmatic execution of a few existing heuristic programs. The library supports a range of common phylogenetic file formats and can be used for both nucleotide and protein data. Furthermore, it is also capable of supporting GPU likelihood calculation on nucleotide character data through the BEAGLE library.

**Availability.** Existing development and source code is available for contribution and for download by the public from GitHub (http://github.com/AlexSafatli/Pylogeny). A stable release of this framework is available for download through PyPi (Python Package Index) at http://pypi.python.org/pypi/pylogeny.

Corresponding author
Alexander Safatli, safatli@cs.dal.ca

## INTRODUCTION

There is a need for tree manipulation, scoring, and flexible heuristic designs as part of larger bioinformatics pipelines. Introduced here is a cross-platform library called Pylogeny intended for heuristic search and analysis of the phylogenetic tree search space, as well as the design of novel algorithms to search this space. This framework is written in the Python programming language, yet it uses efficient auxiliary libraries to perform computationally expensive steps such as scoring. As a programming interface, Pylogeny addresses the needs of both researchers and programmers who are exploring the combinatorial problem associated with phylogenetic trees.

The phylogenetic tree search space describes the combinatorial space of all possible phylogenetic trees for a set of operational taxonomic units. This forms a finite graph

where nodes represent tree solutions and edges represent rearrangement between two trees according to a given operator. Operators include Nearest Neighbor Interchange (NNI), Subtree Prune and Regraft (SPR), and Tree Bisection and Reconnection (TBR), most of which are implemented presently in Pylogeny (*Felsenstein, 2004*). These nodes can be evaluated for fitness against sequence data. We can also evaluate properties of the space such as location of local and global maxima, and the quantity of the latter. The presence of multiple maxima is a confounding factor in heuristic searches which may lead to drawing incorrect conclusions on evolutionary histories.

The source code and library requires only a small number of dependencies. Python dependencies include NumPy (*Walt, Colbert & Varoquaux, 2011*), a ubiquitous numerical library, NetworkX (*Hagberg, Schult & Swart, 2008*), a graph and network library, Pandas (*McKinney, 2010*), a high-performance library for numerical data, and P4 (*Foster, 2004*), a phylogenetic library. An additional dependency that is required is a C phylogenetic library called libpll that underlies the RAxML application and is used to score likelihood of trees (*Stamatakis, 2014*; *Flouri et al., 2014*). Optionally, the BEAGLE (*Ayres et al., 2012*) package could be used for scoring as well. Most dependencies are automatically resolved by a single command by installing the package from the PyPi Package Index. Primary documentation is available on the library's website and alongside the library. All major classes and methods also possess documentation that can be accessed using a native command.

## FEATURES

The functionality to maintain a phylogenetic landscape is implemented in the `landscape` class defined in the `landscape` module of this library. This object interacts with a large number of other classes and supports tree scoring using standard phylogenetic methods. Many of the more relevant objects are tabulated and explained in Table 1. A large coverage of unit testing has been performed on most of the major features including tree rearrangement, heuristic exploration, and landscape construction.

The Pylogeny library can read sequence alignments and character data in formats including FASTA, NEXUS, and PHYLIP. Tree data can only currently be read in a single format with future implementations to allow for a greater breadth of formats. Persistence and management of character data is performed by an alignment module, while trees are stored by their representative string in a tree module. They can be instantiated into a richer topology object in order to manipulate and rearrange them.

### Phylogenetic tree manipulation and scoring

For the purposes of this framework, if instantiated into a `topology` object, phylogenetic trees are modelled in memory as rooted. Therefore, manipulation and access of the tree components, such as nodes and edges, presupposes a rooted structure. Unrooted trees, either multifurcating or bifurcating, can nevertheless still be output and read. Support is also present for splits or bipartitions (as in the `bipartition` object) of these trees, required by many phylogenetic applications such as consensus tree generation (*Margush & McMorris, 1981*).

**Table 1  Overview of the basic objects in the Pylogeny library.**

| Class name | Module name | Description |
| --- | --- | --- |
| alignment | alignment | Represents a biological sequence alignment of character data to enable phylogenetic inference and other operations. |
| treeBranch | base | Represents a branch in a tree structure, such as a phylogenetic tree, and its associated information. |
| treeNode | base | Represents a node in a tree structure, such as a phylogenetic tree, and its associated information. |
| treeStructure | base | A collection of treeNode and treeBranch objects to comprise a tree structure. |
| executable | executable | An interface for the instantation and running of a single call of some given binary application (in a Unix shell). |
| heuristic | heuristic | An interface for a heuristic that explores a state graph and all associated metadata. |
| graph | landscape | Represents a state graph. |
| landscape | landscape | Represents a phylogenetic tree search space, modelled as a graph. |
| vertex | landscape | Represents a single node in the phylogenetic landscape, associated with a tree, and adds convenient functionality to alias parent landscape functionality. |
| landscapeWriter | landscapeWriter | Allows one to write a landscape object to a file, including alignment and tree information. |
| landscapeParser | landscapeWriter | Allows one to parse a landscape that was written to a file. |
| newickParser | newick | Allows one to parse a Newick or New Hampshire format string of characters representing a (phylogenetic) tree. |
| rearrangement | rearrangement | Represents a movement of a branch or node on one tree to another part of that same tree. |
| topology | rearrangement | An immutable representation of a phylogenetic tree where movements can be performed to construct new topology or tree objects. |
| bipartition | tree | Represents a bipartition of a phylogenetic tree. A branch in a phylogenetic tree defines a single bipartition that divides the tree into two disjoint sets of leaves. |
| tree | tree | Represents a phylogenetic tree which does not contain structural information and only defines features such as its Newick string, fitness score, and origin. |
| treeSet | tree | Represents an ordered, disorganized collection of trees that do not necessarily comprise a combinatorial space. |

Iterators can be created for visiting different elements in a tree. Unrooting, rerooting, and other simple manipulation can also be performed on a tree. For more complex manipulation, rearrangement operators (using the rearrangement module) can be performed on a tree to convert it to another topology. To save memory and computation, rearrangements are not performed unless the resultant structure is requested, storing movement information in a transient intermediate structure. This avoids large-scale instantiations of new topology objects when exploring the search space.

Scoring topologies using parsimony or likelihood is done by calling functions present in the library that wrap libpll or the high-performance BEAGLE library. These software packages are written in C or C++, the latter of which allows for increased performance by using the Graphics Processing Unit (GPU) found in a computer for processing.

## Tree search space graph construction and search

The tree search space is abstracted as a graph where a number of graph algorithms and analyses can be performed on it. We do this by utilizing routines found in the NetworkX library which has an efficient implementation of the graph in C. Accessing elements of the graph can be done by iteration or by node name, and properties of the space can be identified by function.

Exploring the space is done by performing rearrangements on trees as `topology` objects where different methods of exploration include a range of enumeration and stochastic-based sampling approaches. In order to make Newick strings consistent amongst trees in a phylogenetic tree search space, an arbitrary but efficient rooting strategy is used to avoid redundancy. Rearranging trees in the search space reroots resultant trees to the lexicographically lowest-order taxa name. This means that different rearrangements that lead to the same topology, with a possibly different ordering of leaves, can still be recognized as not being a new addition to the space. Restriction on this exploration is supported by allowing limitations on movement by disallowing breaking certain bipartitions.

A minimal example to demonstrate constructing a landscape from an alignment file, and finding trees in the space, is found below. The landscape is initialized with a single tree corresponding to an approximate maximum likelihood tree as determined using the FastTree executable (*Price, Dehal & Arkin, 2010*).

```python
from pylogeny.alignment import *
from pylogeny.landscape import landscape

# Open an alignment compatible with the strict
# PHYLIP format.
ali = phylipFriendlyAlignment('al.fasta')
startTree = ali.getApproxMLTree()

# Create the landscape with a root tree.
lscape = landscape(ali,starting_tree=startTree,
                   root=True)

# Explore around that tree.
trees = lscape.exploreTree(lscape.getRoot())
```

The variable `trees` now holds a list of integers. These integers correspond to the names of new trees that have populated our landscape object. These new trees comprise the neighbors of the starting tree, a tree found using FastTree. One could now query the landscape object for new information such as listing these neighbors or writing all of the Newick strings of these trees.

```python
# See trees around the starting tree.
for i in trees: # Iterate over these.
    # Print their Newick strings.
    print lscape.getTree(i)
```

## Applying search space heuristics

Performing a heuristic search of the combinatorial space comprised by a phylogenetic landscape can be done with relative ease using this library. Not only can the heuristic's

steps be later analysed, the resulting space that is explored can also be later viewed and investigated for its properties. The `heuristic` module has a number of already defined approximate methods to discover the global maximum of the space, and with understanding of the object hierarchy, one can create their own.

As an example, one could perform a greedy hill-climbing heuristic on the search space by comparing the trees' parsimony scores. To do this, they would instantiate a `parsimonyGreedy` object from the `heuristic` module and provide an existing landscape and tree in that landscape to initiate the climb. The minimal code to achieve a search from the initial tree would be:

```
from pylogeny.alignment import alignment
from pylogeny.landscape import landscape
from pylogeny.heuristic import parsimonyGreedy

# ali    = Open an alignment file.
# lscape = Construct a landscape.
# ...

h = parsimonyGreedy(lscape,lscape.getRootNode())
h.explore()
```

We have applied a heuristic to the landscape which has populated it with new trees. Nothing is returned here. In order to investigate what new trees have been added, we can query the heuristic object. Furthermore, we can access these new trees from the landscape object.

```
newTrees = h.getPath() # List of tree names.
for name in newTrees:
    # Visit all trees found by heuristic.
    tree = lscape.getTree(name)
    print tree.getScore() # Print scores.
```

## Existing phylogenetic and heuristic programs

The library supports executing other software on its objects. Implementations are present in the framework to call on the FastTree (*Price, Dehal & Arkin, 2010*) and RAxML heuristics for finding an approximate maximum likelihood tree. There is also an implementation for the use of TreePuzzle (*Schmidt et al., 2002*) and CONSEL (*Shimodaira & Hasegawa, 2001*) in order to acquire confidence interval of trees as defined by the Approximately Unbiased (AU) test (*Shimodaira, 2002*). Further implementations can be created by extending a base interface found in the library.

An example of code to demonstrate the use of CONSEL, to generate a confidence interval of trees with default settings, is as follows.

```
from pylogeny.alignment import alignment
from pylogeny.executable import consel

# ali   = Open an alignment file.
# trees = A set of trees (e.g., a landscape).
# ...

AUTest = consel(trees,ali,'AUTestName')
interval = AUTest.getInterval()
```

We now have a `treeSet`, or collection, of tree objects assigned to the variable `interval` which have been deemed to be significant and relevant.

## OTHER LIBRARIES

Other Python libraries that perform similar tasks to this framework include DendroPy (*Sukumaran & Holder, 2010*), ETE, and the P4 Phylogenetic Library. Elements of alignment management and tree manipulation are present in all three of these libraries, but none of them allow for the manipulation and heuristic search of a combinatorial space of phylogenetic trees. There remains a deficiency for this functionality across many languages, Python included. Despite this, this framework can serve to work alongside such libraries for further power.

DendroPy possesses a number of metrics and other tree operations that are not present in Pylogeny. This library supports translating its tree structure to the structure found in DendroPy. Therefore, these operations can still be accessed. Similarly, ETE allows for a number of rich visualization techniques not possible with this framework that can also be accessed in such a manner. A small part of the Pylogeny library is built on the P4 Phylogenetic Library and the P4 library performs a number of operations that are found in this framework, such as scoring and manipulation of trees. We, however, did not find P4 as flexible an API as it appears to be designed for scripting and for work in a terminal rather than as a component of a larger application. For example, P4 likelihood calculations are printed to the standard output rather than returned from a function.

## ACKNOWLEDGEMENTS

The authors thank Professor R. Beiko and the members of Dr. Beiko's Lab in Dalhousie University for some helpful suggestions. The members of the Blouin Lab are also acknowledged for helpful comments and critical review of this manuscript.

### Funding

This work was supported by NSERC Discovery Grant No. 120504858. The funders had no role in study design, data collection and analysis, decision to publish, or preparation of the manuscript.

## Grant Disclosures

The following grant information was disclosed by the authors:
NSERC Discovery: 120504858.

## Competing Interests

The authors declare there are no competing interests.

## Author Contributions

- Alexander Safatli performed the experiments, wrote the paper, prepared figures and/or tables, performed the computation work, reviewed drafts of the paper.
- Christian Blouin reviewed drafts of the paper, provided advice and supervision.

## Data Deposition

The following information was supplied regarding the deposition of related data:
  GitHub: https://github.com/AlexSafatli/Pylogeny.

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
