# Peer review of "Pylogeny: an open-source Python framework for phylogenetic tree reconstruction and search space heuristics"

_PeerJ Computer Science, doi:10.7717/peerj-cs.9_

## Round 0.1 · original submission · Major Revisions

I have now obtained two reviews for your paper and both felt your paper appropriate and the associated software impactful to the field. However, both have provided additional suggestions to both improve the presentation of the software and the usability. Reviewer 2, especially, has gone through an extensive effort to help you improve areas of your presentation. Please consider all these comments very carefully in your revision.

Reviewer 1 ·

Basic reporting

No Comments

Experimental design

No Comments

Validity of the findings

No Comments

Additional comments

Overall, this is a well-written manuscript describing a new python library for phylogenetic tree reconstruction and tree space heuristics. As the authors note at the end of the manuscript, there is a gap in available python libraries to explore tree space and manipulate tree structures. I believe this new resource will be well received by the greater phylogenetics community and is a significant contribution.

The API is documented on the github site and I successfully installed the program and tested some functions. One thing that could be improved in the future is the examples on the github repository. As it stands now there is one example on the repo site; however, there are a few examples in the manuscript. It would be nice if the authors could develop a few simple examples of executed commands and output on the github repository so researchers can test your software and get a feel for the commands. Otherwise, I think the library is currently documented well enough for developers to begin using.

·

Basic reporting

(1) The authors list the dependencies of the package clearly, and this is welcome.
However, as many of these dependencies are independent projects on their own trajectories of development, it is important that the authors include in their documentation (a) the mininum versions of the dependencies required, and (b) the latest version of the dependencies that they have tested. They do the latter with [libpll](http://libpll.org), but it would be useful if the authors did the same for other libraries as well.

(2) I recommend that the authors reference the [libpll](http://libpll.org) library in the abstract, and, perhaps more importantly, very prominantly (i.e., up front in the first few lines) of the README and other documentation. This is important to let users looking for Python bindings that leverage the libpll library quickly and easy locate Pylogeny through basic searches through GitHub or Google (the text describing the libpll usage shows up in the search teaser description).

Experimental design

### Installation

- I recommend that the authors explicitly mention in the program documentation that access to build tools (`gcc` etc.) are required. This may seem very obvious to most of us, but because, sadly, many OS's do install developer tools by default, some new users may be unaware of this, and a simple line in the documentation will save them a lot of frustration.

- I suggest that the author explicitly mention in the program documentation that `setuptools` or `pip` are required for installation, and perhaps provide links in the program documentation for their downloads or commands (e.g., "`sudo apt-get install pip setuptools -y`"). Again, this may seem obvious to anyone who has been doing any Python development work for more than an hour, but it does not do any harm to be as clear and thorough as possible in the documentation (if nothing to reduce the noise the authors will get in terms of spurious bug reports and support requests!).

- I suggest that the authors explicitly mention in the program documentation that access to Python development header include files are required. In Debian/Ubuntu-derived distributions, this can be installed by `sudo apt-get install python-dev -y` (things may be more complicated in OSX). This one, I think, is more tricky for naive users to figure out (the only error is an indication that `Python.h` is missing, and users may not understand that the problem is with their development environment and not Pylogeny). Similarly with `mysqldb`: it is not sufficient for just the package, but developmental headers also seem required.

- For this sort of package, where, the dependency stack is deep and complex, while, at the same time, running on an HPC is a reasonable expectation, it would be immensely useful if the authors could provide instructions on installing Pylogeny in clean environments (i.e., with no assumption of any third-party libraries available) with *no* administrative privileges. I suggest the authors create a clean virtual machine using Vagrant, and attempt to install their library there from scratch without administrative privileges, and present the steps that they take as installation instructions in the README. I recognize that it is impossible to cater to every possible environment, but it should be possible to install in at least one clean environment following the instructions given in the documentation. If the authors can do this, because these instruction are targeted at the most constrained case (clean environment + no administrative privileges), users can easily extrapolate to their own cases.

Validity of the findings

## Major Issues

- The example codes are given, without any explanation of what they really do. In fact, even though the code "works" in the sense that it runs without error, from reading the article text and reading the program documentation alone, it is impossible to understand what the code is supposed to do.

For example, the first example code (page 3) is described as "constructing a landscape from an alignment file, and finding trees in the space". Given the description of the code, I would have expected a collection of tree objects. But instead the return value is a boolean (`True`). Inspecting the `exploreTree` method, that is all that it returns. The documentation for the method claims that it should "Get all neighbors to a tree named i in the landscape using a respective rearrangement operator as defined in the rearrangement module." So, if the method is supposed to return `True`, and something else is meant by "get all neighbors" rather than the tree is to be returned, how do we know what that means? How would users understand how to access those tree returned? The same goes for the second example. `h.explore()` returns `None`. Are the trees loaded in the `h` object? How do we access/view/print those trees? With sufficient experimentation, one can figure it out. But given that the examples are part of a formal publication and stand as canonical reference examples of the library usage, I think they need to be made more complete.

As it happens, the `lscape` object gets populated with the trees in the landscape, and the users need to call `lscape.toTreeSet()`. This may seem obvious to the developers, or even adequately document, but I would argue that this is not the case.
In principle, one could claim that there is sufficient documentation to figure this out (as evidenced, for example, by the fact that I figured it out), but I think this is very poor practice both as a software developer as well as a member of a research community.

Given all this, I think that authors need to expand the working examples to provide end-to-end steps, i.e., going from an alignment file right back out to newick strings. Otherwise the examples are useless (or worse than useless, in that the suggest a full-functional self-contained recipe for a procedure but are sadly incomplete), especially given that the API documentation is similarly lacking. I suggest that the authors create a "cookbook"/"recipe" web page, with *FULL* examples that will run unmodified (i.e., without steps commented out like "#ali = open an alignment file"), and make sure the example demonstrated how to access trees explored. If one of the main features of the library is access to the tree arrangements, then it is not unreasonable to expect that the documentation, article, and examples show at least one complete workflow recovering these tree arrangements.

Note that even if this cookbook/recipe page is done, the examples in the article should still be fleshed out to be more complete (i.e., going end-to-end from alignment to trees).

- I think the major methods should be documented as to what they do, what parameters they take, and what values are returned. I can see that the authors have put in quite a bit of work in this regard, and I really congratulate them on this! However, I think that the documentation needs to be improved before the library can be used by the community. I understand that some perspectives might find documentation such as this as an optional extra, not part of the primary software project. But I feel if the authors have gone to the trouble of writing an article to publish their library (as opposed to simply releasing the library on GitHub), it is an indication that the library is meant to be shared and used by the academic research community. I think documentation of methods in the way I describe would is a mandatory pre-requisite for this. Documentation of the method "contract" ensures that researchers understand exactly how the method is supposed to behave, what inputs it takes, and what they can expect in return. This is, I feel, the difference between a open source project shared with the community and a project that is housed in a publication. These standards are adhered to in, e.g. all R projects. There is no reason Python projects should be allowed to have any sloppier standards of documentation. The authors may wish to adopt the documentation standards of one of their dependencies as an example, and follow the [NumPy standards](https://github.com/numpy/numpy/blob/master/doc/HOWTO_DOCUMENT.rst.txt).

Examples of documentation that needs to be improved:

1.
~~~
__init__(self, dir=None)
x.__init__(...) initializes x; see help(type(x)) for signature Overrides: object.__init__ extit(inherited documentation)
~~~

2.
~~~
addTree(self, tr, score=True, check=True, newick=None, struct=None)
Add a tree to the landscape. Will return its index. Overrides: pylogeny.tree.treeSet.addTree
~~~

3.
~~~
exploreTree(self, i)
Get all neighbors to a tree named i in the landscape using a respective rearrangement operator as defined in the rearrangement module. Rearrangement type is provided as a rearrangement module type definition of form, for example, TYPE_SPR, TYPE_NNI, etc. By default, this is TYPE_SPR.
~~~

The first is simply noise. It should be removed from the public documentation.

The second actually is quite encouraging. But unfortunately fails to describe the remaining parameters.

The third is also almost there: it documents the parameters, but not the return value. Worse, its wording of "Get all neighbors" implied that the "neighbors" are what is returned. Ideally, it should describe itself (for example) as populating the internal set of trees with all neighbors, and that these set of trees can be accessed through such-and-such method, and the that return value is True.

- It is crucial that there is documentation or evidence that the library is functioning as advertised. The library ships with tests, but there is no documentation as to what the tests actually test. The authors do not need to go into great detail about the tests, but the article should reference which aspects of the library have been tested or are covered by unittests (e.g., tree rearrangements, likelihood scoring, etc. etc.).

- The authors mention that Pylogeny supports translating its tree structure to DendroPy. I found no support for this either in the code or the documentation. I personally would find this feature very useful (full disclosure: I am one of the authors of DendroPy), and if the authors are having trouble doing this, I urge them to contact me for assistance. If they have decided not to support this feature, then reference to it being supported should be removed from the article.

Additional comments

- While not critical, unless there is a good reason to do so, why would the authors not follow the standard coding conventions recommended for Python in [PEP-8](https://www.python.org/dev/peps/pep-0008/)? This convention is by no means universally adhered to, and, indeed, even some packages in Python standard library do not follow them. However, the majority of Python packages do, and it is generally recommended.

---

## Round 0.2 · accepted · Accept

Congratulations. Both reviewers found your responses and revisions appropriate. I am now happy to accept your paper for PeerJ. Sorry for the delay in my response, but I had a major NIH grant due this week that has been occupying my time.

Reviewer 1 ·

Basic reporting

No Comments

Experimental design

No Comments

Validity of the findings

No Comments

Additional comments

I want to thank the authors for considering most reviewer comments and suggestions. I specifically want to thank the authors for addressing my comments and adding a working example to the code repository, in addition to contributing more detail to the submodule documentation.

·

Basic reporting

- There are still some errors in the canonical example code (noted below) that should be fixed prior to release/publication. However, despite this, I am still recommending the paper for acceptance, taking it on good faith that the authors will fix these at their earliest convenience in the interests both of the usability of their software as well as benefit to the community.

- First example on the GitHub page (under "Code Example") does not work for me with error:

Traceback (most recent call last):
File "t.py", line 8, in <module>
heu.explore()
File "/home/vagrant/Pylogeny/pylogeny/heuristic.py", line 86, in explore
landscape.exploreTree(cursor['index'])
TypeError: 'NoneType' object has no attribute '__getitem__'

- Example "Rearrangements" has typos (e.g. missing opening quote).

- Example "Heuristics" fails with:

Traceback (most recent call last):
File "t5.py", line 13, in <module>
h.explore()
File "/home/vagrant/Pylogeny/pylogeny/heuristic.py", line 86, in explore
landscape.exploreTree(cursor['index'])
TypeError: 'NoneType' object has no attribute '__getitem__'

- DendroPy conversion: this works by serializing the a Pylogeny tree instance to a Newick string and then re-reading the tree using DendroPy. For most purposes, this should be sufficient (though the current implementation does not support control of, e.g., edge lengths, rooting state etc.). Helpful hint: ``Tree.read_from_string()`` is no longer supported as of DendroPy 4.xx onwards. Instead of:

t = dendropyTree()
t.read_from_string(self.toNewick(),'newick')

use:

t = dendropyTree.get_from_string(
self.toNewick(),
schema='newick')

or, using the more modern:

t = dendropyTree.get(
data=self.toNewick(),
schema='newick')

- With respect to recommendations made in the original review, the authors have indeed revised the documentation and examples, as well as installation instructions. In particular, the detailed annotatations accompanying the code exampels in the main paper itself, as well as the table describing the structure and outline of the library, is very welcome! Kudos!

- The primary author states in the rebuttal letter that: "I have chosen to, for the time being, dismiss the suggestion to follow the standard coding conventions recommended for Python as this is something planned for a future milestone and would take a considerable amount of time for relatively small reward. So, while the authors promise this will be done in a future release, upon resubmission this has not been done yet". I think it is fine not (though regrettable) not to follow the Python coding conventions. However, I think once the software is released, the API should not be changed just to conform to coding conventions.

Experimental design

(All remarks are in the "Basic Reporting" field)

Validity of the findings

(All remarks are in the "Basic Reporting" field)

Additional comments

(All remarks are in the "Basic Reporting" field)